# High spatial resolution analysis using automated indentation mapping differentiates biomechanical properties of normal vs. degenerated articular cartilage in mice

**Anand O Masson[1,2], Bryce Besler[1,2], W Brent Edwards[1,2,3], Roman J Krawetz[1,2,4,5]***

[1]Biomedical Engineering Graduate Program, University of Calgary, Calgary, Canada; [2]McCaig Institute for Bone and Joint Health, University of Calgary, Calgary, Canada; [3]Human Performance Laboratory, Faculty of Kinesiology, University of Calgary, Calgary, Canada; [4]Department of Surgery, Cumming School of Medicine, University of Calgary, Calgary, Canada; [5]Department of Cell Biology and Anatomy, Cumming School of Medicine, University of Calgary, Calgary, Canada

*For correspondence:
rkrawetz@ucalgary.ca

Competing interest: The authors declare that no competing interests exist.

**Abstract** Characterizing the biomechanical properties of articular cartilage is crucial to understanding processes of tissue homeostasis vs. degeneration. In mouse models, however, limitations are imposed by their small joint size and thin cartilage surfaces. Here we present a three-dimensional (3D) automated surface mapping system and methodology that allows for mechanical characterization of mouse cartilage with high spatial resolution. We performed repeated indentation mappings, followed by cartilage thickness measurement via needle probing, at 31 predefined positions distributed over the medial and lateral femoral condyles of healthy mice. High-resolution 3D x-ray microscopy (XRM) imaging was used to validate tissue thickness measurements. The automated indentation mapping was reproducible, and needle probing yielded cartilage thicknesses comparable to XRM imaging. When comparing healthy vs. degenerated cartilage, topographical variations in biomechanics were identified, with altered thickness and stiffness (instantaneous modulus) across condyles and within anteroposterior sub-regions. This quantitative technique comprehensively characterized cartilage function in mice femoral condyle cartilage. Hence, it has the potential to improve our understanding of tissue structure-function interplay in mouse models of repair and disease.

## Editor's evaluation

The manuscript provides supportive evidence for high-resolution analysis using indentation mapping that differentiates the biomechanical properties of normal vs. degenerated mouse articular cartilage. This study will provide the basis for future analyses for assessing degenerated articular cartilage in different mouse models (genetically engineered or assessing the therapeutic targets).

## Introduction

The articular cartilage in synovial joints has a specialized three-dimensional (3D) structure and biochemical composition that provide low-friction, wear-resistance, and load-bearing properties to the tissue. Mechanical loading is essential to tissue homeostasis and influences gene expression, chondrocyte metabolism, extracellular matrix maintenance, and associated interstitial fluid permeability (**Vincent**

*and Wann, 2019*; *Grodzinsky et al., 2000*). Therefore, the biomechanical properties of cartilage are unique indicators of tissue homeostasis versus degeneration. Indeed, research into cartilage-related changes during the development of degenerative diseases such as osteoarthritis (OA) has demonstrated that early tissue dysfunction involves loss of proteoglycans and increased water content, resulting in reduced compressive strength and higher tissue permeability (*Stolz et al., 2009*; *Setton et al., 1994*). Disease progression leads to further structural damage and altered mechanics (*Kleemann et al., 2005*), ultimately resulting in tissue failure.

Mouse models are commonly used in pre-clinical studies focused on cartilage repair and pathophysiology of OA (*McCoy, 2015*). Mice are advantageous because of the well-established development of genetically modified strains (*Helminen et al., 2002*) and the availability of diverse model systems mimicking mechanisms of spontaneous and induced tissue degeneration and regeneration (*Bedelbaeva et al., 2010*; *Kyostio-Moore et al., 2011*; *Murphy et al., 2020*; *Christiansen et al., 2015*). In this regard, mouse models serve as powerful tools for the targeted assessment of cellular and molecular processes, and the discovery of novel therapeutics related to cartilage regeneration and OA. Yet, while the structural integrity and biochemical composition of murine cartilage are routinely assessed through histological and molecular approaches, the evaluation of how these features translate into mechanical function is limited. The main challenge in mechanical function assessment stems from the small joint size and thin cartilage found in mice relative to other species (*Malda et al., 2013*). Prior efforts to overcome these challenges include finite element modeling and optimization of small-scale indentation techniques (*Stolz et al., 2009*; *Cao et al., 2006*; *Berteau et al., 2016*; *Das Neves Borges et al., 2014*).

Mechanical indentation (including atomic force microscopy [AFM] techniques), performed using either creep or stress-relaxation protocols, is widely employed for assessing the biomechanical behavior of cartilage and OA-related changes in many species (*Athanasiou et al., 1991*; *Risch et al., 2021*; *Hoch et al., 1983*) and considered the gold standard for small animal joints (*Lakin et al., 2017*). Unlike confined and unconfined compression tests, no sectioning or subsampling of tissue (i.e. cylindrical explants) is required (*Mansour, 2013*). Instead, the cartilage tissue and its subchondral bone interface is kept intact, providing more physiologically relevant data. Indentation is also advantageous in that it is non-destructive and allows for repeated measures in situ (*Lu and Mow, 2008*). However, the natural curvature of joint articular surfaces poses a challenge to testing, as indentation must be conducted perpendicularly to the surface (*Swann and Seedhom, 1989*).

Recently, a novel automated indentation technique has been developed for the mechanical assessment of cartilage (*Sim et al., 2017*). This commercially available multi-axial apparatus (Mach-1, *Biomomentum Inc,* Laval, QC) is capable of detecting specimen surface orientation at each position of measurement and subsequently indent normal to the surface (*Sim et al., 2017*; *Biomomentum, 2020*). As such, it can map entire cartilage surfaces using a single setup with high spatial density and has been previously shown to discriminate between healthy and diseased human cartilage samples (*Sim et al., 2017*; *Seidenstuecker et al., 2019*). This apparatus has been recently employed by Woods and colleagues (*Woods et al., 2021*) to evaluate altered biomechanics in a mouse model of cartilage degeneration. However, the study was conducted in the non-load-bearing region of the mouse knee joint. Considering that the knee range of motion in mice is between 40 and 145° (i.e. unable to fully extend) (*Jia et al., 2018*), with normal gait range between 90.5 and 120° (extension-flexion) (*Hu et al., 2017*), the contact regions in the distal femoral condyles are located further posteriorly. Careful consideration of species-specific differences in knee joint anatomy and kinematics is imperative for proper translation of pre-clinical models (*Oláh et al., 2021*).

Woods and colleagues (*Woods et al., 2021*) were also unable to account for site-specific cartilage thickness variations in their measurements, instead of using the mean cartilage thickness obtained via histological analysis in their assessment (*Woods et al., 2021*). Other studies have employed mean cartilage thickness values retrieved through histology or imaging techniques to characterize and model tissue mechanical parameters in mice (*Cao et al., 2006*; *Das Neves Borges et al., 2014*). Implicit in this approach is the underlying assumption that cartilage thickness is relatively uniform among medial and lateral compartments (e.g. femoral condyles or tibial plateaus), and along their anteroposterior or mediolateral axis. Yet, regional variations in thickness are recognized within these cartilage surfaces (*Malda et al., 2013*; *Das Neves Borges et al., 2014*; *Li et al., 2005*) and known to impact small-scale indentation measurements (*Swann and Seedhom, 1989*; *Hayes et al., 1972*).

Hence, the purpose of this study was to investigate the reliability of automated indentation mapping in the assessment of healthy femoral articular cartilage in mice and characterize site-specific variations in cartilage thickness. We also employed concurrent contrast-enhanced 3D x-ray microscopy (XRM) imaging to validate the cartilage thickness measurements from our optimized needle probing (NP) protocol (*Jurvelin et al., 1995*). Finally, this approach was used to investigate biomechanical differences in a clinically relevant mouse model of cartilage degeneration (*Rhee et al., 2005*). Together, we show that automated indentation is reliable and able to characterize topographic and mechanical variations across condyle cartilage locations in intact cartilage. Moreover, this technique was able to identify regional changes in cartilage thickness and stiffness in degenerated cartilage. A comprehensive and standardized biomechanical evaluation of cartilage in repair and disease can greatly contribute to our understanding of tissue structure-function interplay, thereby enhancing the clinical relevance of mouse models in this area.

## Results

### Automated indentation mapping reliability

While previous studies have employed indentation mapping on mouse cartilage, to our knowledge none has reported on its precision and test-retest reliability. Therefore, we performed three repeated mappings of 31 predefined positions distributed over the femoral condyles of 10 C57BL/6 mice (*Figure 1A*) to assess the reliability of the automated surface indentation technique. The setup was developed and optimized (*Figure 1—figure supplement 1*) to assess the load-bearing regions of the femoral condyles and achieve non-destructive retrieval of specimen post-testing for subsequent 3D XRM imaging analysis. The imposed step deformation (i.e. indentation depth) on the femoral cartilage yielded typical stress-relaxation behavior, characterized by a sharp increase in force followed by gradual relaxation over time until equilibrium (*Figure 1B*). Assessment of stress-relaxation and corresponding force-displacement curves (*Figure 1B*) demonstrated consistency among repeated measurements for single positions and visible differences in peak reaction forces between condyles. These observations were further evidenced by the spatial distribution of peak force values across condylar testing sites (*Figure 1C*, *Table 1*). A total of 930 indentation measurements were retrieved, out of which only 21 produced atypical curves (7 testing sites at specimen's periphery with higher angles yielded noisy signals, *Figure 1C*), representing a 2.26% error rate during data acquisition. High reliability and absolute agreement between repeated measures for individual testing sites were observed, with 4.7% intra-assay average coefficient of variation (*Table 1*) and intraclass correlation coefficients – ICC (lower 95%, upper 95%) – ranging from 0.974 (0.966, 0.981) for the lateral condyle to 0.971 (0.963, 0.978) for the medial condyle. Mean peak force values illustrate site-specific variations within and between condyles (*Figure 1D*). The lateral condyle values varied significantly per position (p<0.0001), ranging from 0.07 to 0.15 N and showed a trend for higher values at outermost positions, with a slight decrease in force posteriorly. The latter was also seen for the medial condyle, wherein heterogeneities in peak force were apparent (p<0.0001) and had a wider range – from 0.15 to 0.294 N. Since the analysis per testing site also reflects inherent deviations due to anatomical positioning across specimens, data was pooled for regional (between condyles) and sub-regional (between and within anteroposterior locations) comparisons.

As *Table 2* shows, the average peak force was significantly higher on the medial condyle and on both its anterior and posterior sub-regions when compared to lateral counterparts (Lat/Ant vs. Med/Ant and Lat/Post vs. Med/Post). Interestingly, no significant differences were observed between sub-regions of the lateral condyle (Lat/Ant vs. Lat/Post). In contrast, the mean peak force yielded at the Med/Post sub-region was 20% lower than on the Med/Ant (p<0.01). As cartilage thickness variations between and within condyle locations could affect peak forces measured at same indentation depth (*Michalak et al., 2019*), with thinner cartilage yielding higher force values, we sought to determine the cartilage thickness distribution within the same surfaces and validate this approach using XRM imaging.

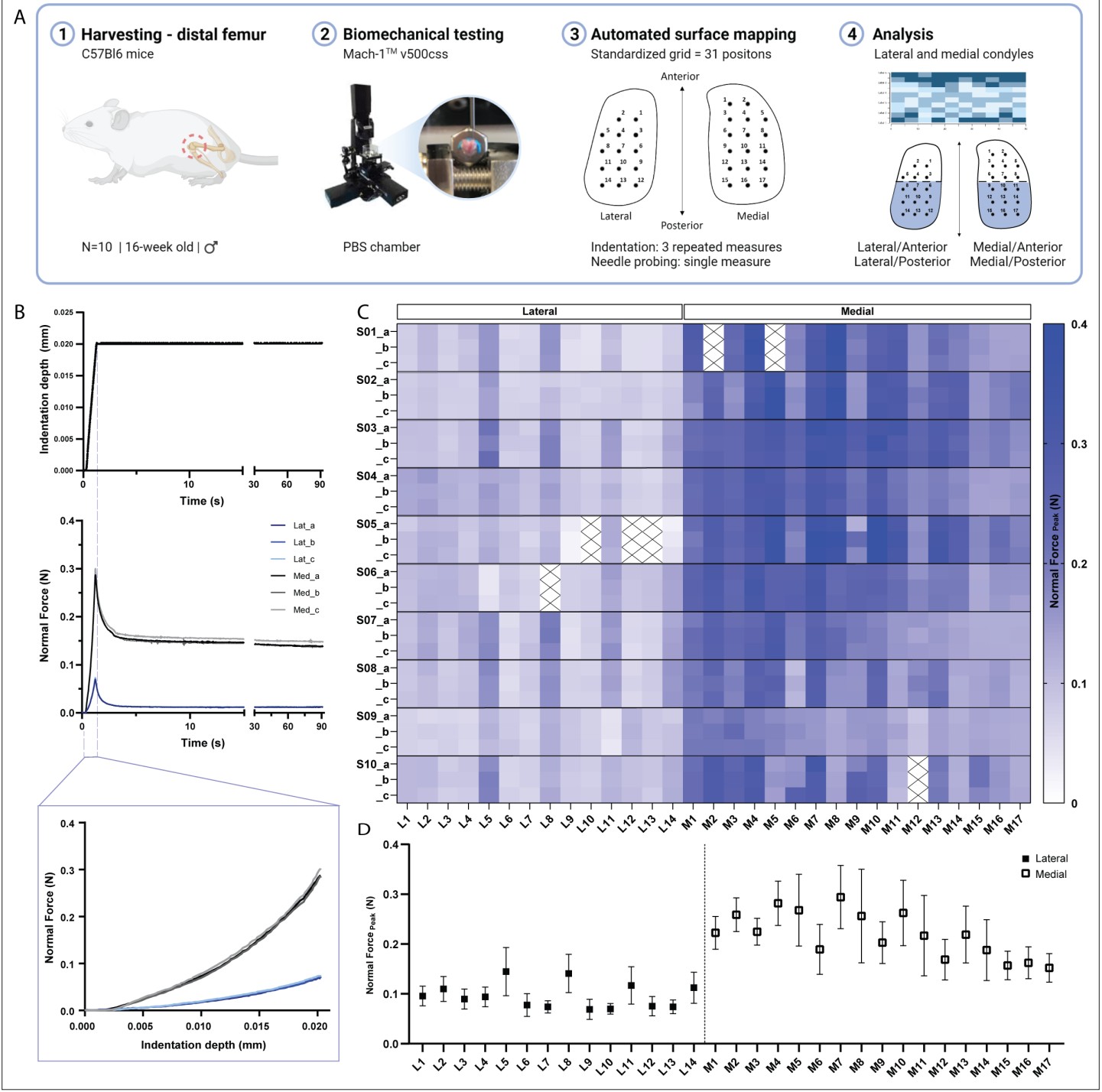

**Figure 1.** Indentation mapping of murine articular cartilage. (**A**) Schematic overview of experimental design employed for biomechanical testing of murine articular cartilage using Mach-1 v500css mechanical tester. (**B**) Step displacement used for cartilage indentation (top), with typical force-relaxation response curves obtained for three repeated measures on representative lateral and medial condyle positions (middle) and corresponding force increase with indentation depth (bottom). (**C**) Normal peak force recorded for all three repeated measures (**A-C**), considering each of the 31 testing sites, L1-L14 at lateral condyle and M1-M17 at medial condyle, for each specimen (n=10), (S01–S10), demonstrates general agreement for intra-specimen measurements on both condyles. (**D**) Mean peak force values varied within and between condyle locations and higher within medial condyle testing sites. Data is presented as mean ± SD.

The online version of this article includes the following figure supplement(s) for figure 1:

**Figure supplement 1.** Specimen preparation and assemble to sample holder for automated indentation mapping using custom setup, allowing for repositioning of the sample and non-destructive retrieval for three-dimensional x-ray microscopy imaging (scale bars equal to 4 mm).

**Table 1.** Mean peak force ($\bar{X}$:N) and coefficient of variation (CV:%) from triplicate measurements for each of the 31 positions (L1-M17) of assessment over mice femoral condyles.

| | S01 | | S02 | | S03 | | S04 | | S05 | | S06 | | S07 | | S08 | | S09 | | S10 | |
|---|---|---|---|---|---|---|---|---|---|---|---|---|---|---|---|---|---|---|---|---|
| | $\bar{X}$ | CV | $\bar{X}$ | CV | $\bar{X}$ | CV | $\bar{X}$ | CV | $\bar{X}$ | CV | $\bar{X}$ | CV | $\bar{X}$ | CV | $\bar{X}$ | CV | $\bar{X}$ | CV | $\bar{X}$ | CV |
| L1 | 0.08 | 2.9 | 0.08 | 3.0 | 0.10 | 4.1 | 0.13 | 1.8 | 0.09 | 12.1 | 0.10 | 2.7 | 0.10 | 4.9 | 0.09 | 3.3 | 0.07 | 1.5 | 0.11 | 0.2 |
| L2 | 0.11 | 1.3 | 0.09 | 2.5 | 0.10 | 2.3 | 0.14 | 6.1 | 0.12 | 8.2 | 0.11 | 4.5 | 0.13 | 4.9 | 0.11 | 6.1 | 0.07 | 13.7 | 0.11 | 5.7 |
| L3 | 0.07 | 4.9 | 0.08 | 3.6 | 0.08 | 5.0 | 0.12 | 2.5 | 0.11 | 4.8 | 0.12 | 1.5 | 0.09 | 3.8 | 0.08 | 4.9 | 0.07 | 8.5 | 0.08 | 8.0 |
| L4 | 0.10 | 7.2 | 0.07 | 2.8 | 0.10 | 1.3 | 0.11 | 1.6 | 0.08 | 3.6 | 0.11 | 4.7 | 0.13 | 2.2 | 0.09 | 3.0 | 0.06 | 4.8 | 0.09 | 5.5 |
| L5 | 0.16 | 0.7 | 0.15 | 5.0 | 0.21 | 9.0 | 0.14 | 7.7 | 0.12 | 7.8 | 0.04 | 10.0 | 0.17 | 2.4 | 0.17 | 4.2 | 0.11 | 6.0 | 0.19 | 6.3 |
| L6 | 0.06 | 9.3 | 0.07 | 7.4 | 0.08 | 13.4 | 0.12 | 4.1 | 0.09 | 5.7 | 0.08 | 2.9 | 0.06 | 9.0 | 0.06 | 6.9 | 0.08 | 6.0 | 0.07 | 5.8 |
| L7 | 0.06 | 4.7 | 0.07 | 6.2 | 0.08 | 6.9 | 0.10 | 3.1 | 0.07 | 12.3 | 0.07 | 3.8 | 0.09 | 8.0 | 0.08 | 4.9 | 0.07 | 3.8 | 0.07 | 7.5 |
| L8 | 0.16 | 2.2 | 0.09 | 15.9 | 0.19 | 3.2 | 0.14 | 4.9 | 0.11 | 7.9 | - | - | 0.20 | 5.8 | 0.17 | 3.1 | 0.12 | 3.5 | 0.12 | 6.5 |
| L9 | 0.05 | 10.3 | 0.07 | 5.4 | 0.08 | 2.4 | 0.09 | 1.5 | 0.03 | 20.2 | 0.07 | 0.3 | 0.05 | 6.4 | 0.07 | 2.9 | 0.09 | 0.9 | 0.08 | 2.5 |
| L10 | 0.04 | 4.2 | 0.07 | 6.8 | 0.07 | 3.1 | 0.08 | 4.5 | - | - | 0.08 | 3.3 | 0.07 | 3.9 | 0.07 | 5.5 | 0.07 | 1.1 | 0.08 | 3.2 |
| L11 | 0.10 | 5.7 | 0.08 | 6.1 | 0.15 | 2.4 | 0.11 | 9.1 | 0.13 | 5.1 | 0.14 | 6.8 | 0.16 | 5.9 | 0.14 | 3.7 | 0.04 | 9.0 | 0.11 | 1.3 |
| L12 | 0.05 | 0.9 | 0.08 | 4.6 | 0.07 | 9.4 | 0.07 | 3.6 | - | - | 0.09 | 3.1 | 0.06 | 6.2 | 0.06 | 8.8 | 0.11 | 3.6 | 0.10 | 3.3 |
| L13 | 0.05 | 3.0 | 0.07 | 3.6 | 0.07 | 6.4 | 0.07 | 5.8 | - | - | 0.10 | 2.3 | 0.07 | 2.9 | 0.08 | 4.9 | 0.09 | 0.9 | 0.09 | 6.0 |
| L14 | 0.08 | 5.0 | 0.08 | 5.7 | 0.13 | 4.9 | 0.10 | 4.9 | 0.04 | 19.0 | 0.14 | 1.1 | 0.14 | 3.7 | 0.16 | 1.7 | 0.09 | 12.0 | 0.12 | 1.4 |
| M1 | 0.29 | 3.7 | 0.18 | 6.9 | 0.23 | 4.1 | 0.23 | 1.9 | 0.21 | 5.5 | 0.26 | 1.4 | 0.22 | 6.7 | 0.20 | 2.2 | 0.20 | 1.7 | 0.22 | 1.9 |
| M2 | - | - | 0.25 | 4.6 | 0.25 | 3.2 | 0.26 | 1.0 | 0.28 | 2.3 | 0.29 | 2.1 | 0.27 | 1.5 | 0.24 | 6.3 | 0.20 | 5.4 | 0.29 | 0.9 |
| M3 | 0.23 | 3.1 | 0.17 | 3.8 | 0.24 | 0.7 | 0.24 | 1.4 | 0.24 | 2.8 | 0.24 | 1.8 | 0.23 | 3.2 | 0.19 | 2.7 | 0.22 | 6.7 | 0.25 | 3.4 |
| M4 | 0.36 | 5.0 | 0.30 | 6.5 | 0.28 | 0.6 | 0.30 | 2.6 | 0.29 | 3.1 | 0.31 | 2.7 | 0.25 | 5.8 | 0.26 | 3.1 | 0.19 | 4.5 | 0.30 | 2.7 |
| M5 | - | - | 0.38 | 10.0 | 0.30 | 3.9 | 0.30 | 1.9 | 0.36 | 1.1 | 0.26 | 0.7 | 0.30 | 2.4 | 0.25 | 5.5 | 0.18 | 4.3 | 0.15 | 5.1 |
| M6 | 0.19 | 1.7 | 0.17 | 6.3 | 0.25 | 3.4 | 0.23 | 3.8 | 0.19 | 5.6 | 0.26 | 2.3 | 0.21 | 0.7 | 0.11 | 10.9 | 0.16 | 3.9 | 0.20 | 15.1 |
| M7 | 0.30 | 3.3 | 0.30 | 4.2 | 0.34 | 5.9 | 0.30 | 1.9 | 0.37 | 7.0 | 0.31 | 1.7 | 0.26 | 2.2 | 0.28 | 1.1 | 0.16 | 1.4 | 0.30 | 6.9 |
| M8 | 0.37 | 3.3 | 0.36 | 6.2 | 0.33 | 2.0 | 0.28 | 5.1 | 0.35 | 5.6 | 0.23 | 2.0 | 0.23 | 6.0 | 0.17 | 3.6 | 0.13 | 1.7 | 0.14 | 4.2 |
| M9 | 0.20 | 3.8 | 0.16 | 2.5 | 0.27 | 5.6 | 0.21 | 2.8 | 0.18 | 19.6 | 0.25 | 1.2 | 0.20 | 4.2 | 0.16 | 0.9 | 0.15 | 3.4 | 0.24 | 18.6 |
| M10 | 0.26 | 2.8 | 0.30 | 2.0 | 0.33 | 7.1 | 0.28 | 1.2 | 0.39 | 2.2 | 0.25 | 2.4 | 0.19 | 2.5 | 0.25 | 2.0 | 0.16 | 2.4 | 0.24 | 2.7 |
| M11 | 0.28 | 2.0 | 0.30 | 2.8 | 0.30 | 6.0 | 0.21 | 1.6 | 0.31 | 3.8 | 0.22 | 2.0 | 0.16 | 1.1 | 0.14 | 0.7 | 0.13 | 7.8 | 0.11 | 7.1 |
| M12 | 0.16 | 1.6 | 0.19 | 5.9 | 0.25 | 1.6 | 0.18 | 0.6 | 0.17 | 2.3 | 0.18 | 6.4 | 0.15 | 6.9 | 0.18 | 5.7 | 0.09 | 7.7 | - | - |
| M13 | 0.22 | 4.3 | 0.26 | 4.6 | 0.29 | 3.0 | 0.20 | 2.3 | 0.32 | 4.0 | 0.20 | 2.6 | 0.15 | 1.2 | 0.21 | 3.6 | 0.14 | 0.1 | 0.20 | 2.9 |
| M14 | 0.19 | 1.6 | 0.27 | 1.2 | 0.25 | 4.2 | 0.17 | 2.6 | 0.27 | 1.2 | 0.20 | 0.8 | 0.14 | 2.9 | 0.13 | 4.1 | 0.12 | 3.2 | 0.12 | 1.8 |
| M15 | 0.16 | 3.7 | 0.16 | 0.5 | 0.18 | 1.6 | 0.14 | 1.6 | 0.18 | 0.7 | 0.13 | 1.9 | 0.13 | 1.7 | 0.14 | 1.1 | 0.14 | 1.6 | 0.21 | 9.2 |
| M16 | 0.14 | 1.3 | 0.19 | 1.3 | 0.20 | 0.9 | 0.14 | 2.1 | 0.22 | 1.4 | 0.16 | 2.7 | 0.12 | 0.8 | 0.15 | 1.2 | 0.13 | 1.4 | 0.16 | 1.7 |
| M17 | 0.14 | 2.2 | 0.21 | 1.1 | 0.17 | 3.4 | 0.15 | 2.6 | 0.19 | 3.1 | 0.15 | 5.5 | 0.13 | 2.1 | 0.12 | 3.0 | 0.13 | 1.4 | 0.13 | 2.2 |

## Cartilage thickness characterization: comparison between needle probing and XRM imaging

NP thickness mapping was performed on all 10 femoral condyles, which were subsequently scanned using contrast enhanced XRM imaging (*Figure 2—figure supplement 1*). As shown in *Figure 2A*, reconstructed 3D datasets of murine distal femurs allowed us to validate the spatial distribution of NP testing sites, whereby the corresponding region of interest (ROI) coordinates for imaging processing could be determined. Additionally, the two-dimensional slices confirmed that the needle probe pierced

**Table 2.** Mean and standard deviation (SD) values for peak force (N) as determined by automated indentation test performed for n=10 distal femur samples of murine articular cartilage.
Mean values compared between condyles (Lateral/Medial; unpaired, Student's *t* test, *α*=0.05) and within sub-regions of condyles (Lat/Ant, Lat/Post, Med/Ant, Med/Post; *one-way* ANOVA, p<0.05).

| | Peak force [N] | | | |
|---|---|---|---|---|
| Condyle | Mean (SD) | Condyle | Mean (SD) | Lat - Med |
| Lateral | 0.096 (0.011) | Medial | 0.219 (0.037) | p<0.0001 |
| *Lat/Ant* | 0.107 (0.017) | *Med/Ant* | 0.250 (0.037) | p<0.0001 |
| *Lat/Post* | 0.090 (0.011) | *Med/Post* | 0.199 (0.037) | p<0.0001 |
| Ant - Post | *ns* | Ant - Post | p<0.01 | |

ns, no significant difference.
-, comparison not applicable.

the full length of the cartilage, reaching the subchondral bone (*Figure 2A*). The cartilage surface and cartilage-subchondral bone interface positions were identified using the load-displacement curves from NP (*Figure 2B*), and the cartilage thickness for each position was then calculated considering the surface angle (see Methods section for details). There was a 2.58% rate of needle probe failure (8 out of 310 measurements, testing sites near the specimen's edge) during data acquisition. NP and contrast enhanced XRM imaging yielded similar cartilage thickness distributions on both condyles (*Figure 2C*), demonstrating a highly significant correlation between paired values (R=0.842, n=302, p<0.0001) (*Figure 2D*). Further method-comparison using a Bland-Altman plot (*Figure 2E*) illustrated XRM measurements were approximately 6.8 µm thicker on average than NP, representing only a 1.55 voxels difference (4.39 µm resolution - XRM). Moreover, no significant differences were found between pairwise mean thickness values (NP vs. XRM) for individual positions within the lateral condyle (*Figure 2F*). Similar results were shown for the medial condyle, wherein higher mean thickness values from XRM were only seen on M06 (Δ=8.68 µm, p=0.028), M11 (Δ=9.89 µm, p<0.01), and M13 (Δ=9.34 µm, p=0.012) testing sites. No obvious reason for these localized differences was found, and dullness of needle was ruled out since subsequent testing sites (M14 to M17) yielded comparable cartilage thickness values between techniques.

Unlike peak force distributions, spatial heterogeneities in cartilage thickness were apparent among individual testing sites only on the lateral condyle (p<0.0001), with averaged values ranging from 46 to 76 µm; whereas thickness distributions within the medial condyle were more uniform (p=0.06), ranging from 36 to 45 µm. Nevertheless, no significant differences were observed within condyles when comparing their anteroposterior sub-regions; whereas the medial condyle was significantly thinner than its lateral counterpart, both in its anterior and posterior sub-regions (*Tables 3 and 4*). XRM segmentation of contralateral femoral condyles (n=5) not subjected to biomechanical testing (*Figure 2—figure supplement 2*) yielded mean cartilage thicknesses of 40.63±3.14 µm and 56.99±6.26 µm for the medial and lateral condyles, respectively. Comparison to the averaged values retrieved by site-specific NP and XRM following NP suggests repeated indentations did not compromise cartilage before NP test, whereas compromise of cartilage integrity by NP may have impacted subsequent XRM evaluation leading to overestimation of site-specific cartilage thicknesses.

As expected, pairwise comparisons for individual testing sites demonstrated a significant negative correlation between peak force and thickness measurements within lateral (R=−0.554, *n*=135, p<0.001, *Figure 3A*) and medial (R=−0.463, *n*=167, p<0.001, *Figure 3B*) condyles. Knowledge of site-specific thickness variations allowed the compressive stiffness to be determined for the same mechanical strain at each testing site. Instantaneous modulus was calculated using Hayes et al. elastic model (*Hayes et al., 1972*) at 20% strain, wherein linear elastic behavior can be assumed, and instantaneous response is considered as flow-independent (*Poisson's* ratio, $\upsilon$ =0.5 assumed; *Armstrong, 1986*). Regardless of femoral condyle, instantaneous stiffness demonstrated no significant correlation to thickness variations (*Figure 3C–D*). Notably, compressive stiffness differed significantly between condyles (Lat vs. Med, p<0.001), but no longer within a condyle (Lat/Ant vs Lat/Post: p>0.99; Med/Ant vs Med/Post: p=0.546), like seen for peak reaction force on the medial side. Together, cartilage

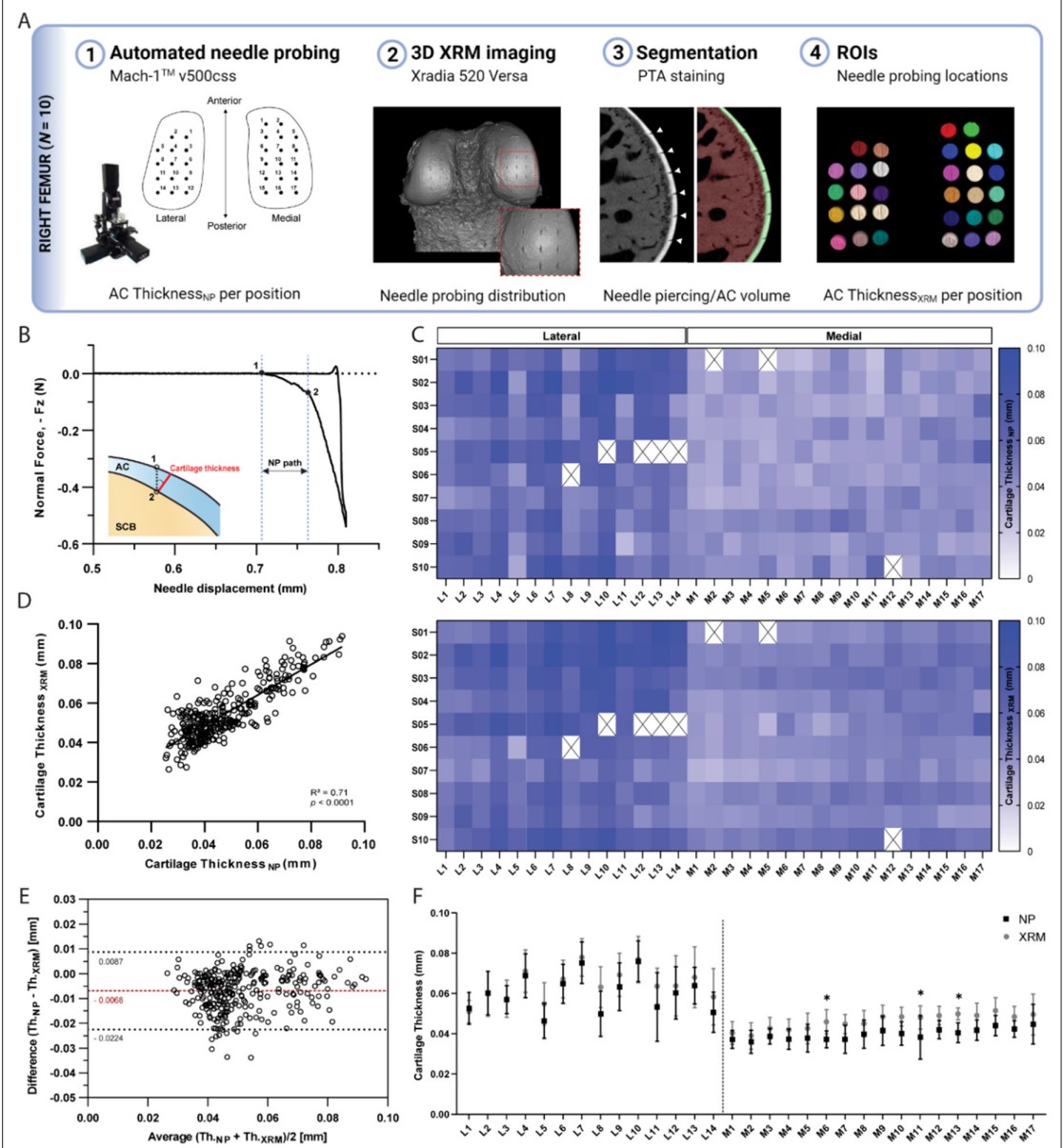

**Figure 2.** Thickness mapping of murine articular cartilage. (**A**) Assessment of agreement between needle probing (NP) and x-ray microscopy (XRM) imaging cartilage thickness per position of measurement within right femoral condyles (n=10). (**B**) Representative normal force-displacement curve obtained during NP test depicting articular cartilage (AC) surface (1) and subchondral bone interface (2) positions and cartilage thickness calculated normal to the surface (*red*) using the surface angle orientation. (**C**) Mapping distributions of cartilage thickness values per position as measured by NP and XRM. (**D**) Correlation graph of cartilage thickness measured by NP vs. XRM, *R*=0.842, n=302, p<0.0001, and corresponding (**E**) Bland-Altman plot showing overall agreement between methods, with average difference of 6.8 µm in thickness. Dotted black lines show upper and lower 95% limit of agreement. (**F**) Pairwise assessment of mean cartilage thickness NP vs. XRM per position for the lateral and medial condyles (* p<0.05, ** p<0.01, *two-way* ANOVA). Symbols represent the means and error bars the standard deviation.

The online version of this article includes the following source data and figure supplement(s) for figure 2:

**Source data 1.** Needle probe thickness measurements for *Figures 2 and 4*.

**Figure supplement 1.** Setup for needle probing thickness measurement, using a 30G×1.4" hypodermic needle (TSK Laboratory, Japan) adapted to the 1-mm spherical indenter (Biomomentum Inc, Laval, QC)(scale bars equal to 4 mm).

**Figure supplement 2.** XRM imaging of femoral condyles.

**Table 3.** Mean and standard deviation (SD) values for cartilage thickness as determined by needle probing for distal femur samples of murine articular cartilage (n=10).
Mean values compared between condyles (Lateral/Medial; unpaired, Student's *t* test) and within subregions of condyles (Lat/Ant, Lat/Post, Med/Ant, Med/Post; *one-way* ANOVA, $\alpha$=0.05).

| Cartilage thickness [μm] | | | | |
|---|---|---|---|---|
| **Condyle** | **Mean (SD)** | **Condyle** | **Mean (SD)** | **Lat - Med** |
| Lateral | 60.3 (6.3) | Medial | 39.8 (2.9) | p<0.0001 |
| *Lat/Ant* | 57.0 (5.9) | *Med/Ant* | 37.5 (3.8) | p<0.0001 |
| *Lat/Post* | 62.3 (7.8) | *Med/Post* | 41.9 (3.0) | p<0.0001 |
| Ant - Post | *ns* | Ant - Post | *ns* | |

ns, no significant difference.
-, comparison not applicable.

thickness appears as a contributing factor but not the sole explanation to mechanical variations, which is also affected by differences in composition and morphology. Next, we assessed the potential of this indentation testing in identifying microscale biomechanical differences between healthy and degraded cartilage.

## Altered biomechanical properties in degenerated murine articular cartilage

To assess the changes in mechanical response within the context of cartilage degeneration, we employed the same testing protocol on age-matched Proteoglycan 4 (PRG4) knockout mice (n=6) and compared the outcomes to the C57BL/6 controls. PRG4 is a mucin-like glycoprotein highly conserved across species (*Askary et al., 2016*; *Ikegawa et al., 2000*) and functionally relevant in joint homeostasis and lubrication (*Rhee et al., 2005*; *Jay et al., 2007*; *Coles et al., 2010*). PRG4 loss of function, as seen in knockout mice (*Prg4$^{-/-}$*), leads to degenerative joint changes recapitulating the phenotype of human camptodactyly-arthropathy-coxavara-pericarditis syndrome (*Rhee et al., 2005*; *Marcelino et al., 1999*). As a spontaneous model of cartilage degeneration, *Prg4$^{-/-}$* structural and morphological joint changes are expected to be more consistent across specimens and less susceptible to variability derived from models that rely on injury/intervention. Histological alterations of articular cartilage have been comprehensively described and include surface roughness, tissue thickening, and loss of collagen parallel orientation at the superficial layer, progressing to irreversible tissue damage with age (*Rhee et al., 2005*; *Karamchedu et al., 2016*). Yet, the microscale assessment of site-specific mechanical variations, aside from friction, has not been described so far for *Prg4$^{-/-}$* knee cartilage surfaces. Mapping of biomechanical parameters allowed site-specific differences in *Prg4$^{-/-}$* cartilage to be visualized, and outcomes were largely reproducible across femoral specimens (*Figure 4A–B*).

**Table 4.** Mean and standard deviation (SD) values for cartilage thickness as determined by x-ray microscopy imaging for distal femur samples of murine articular cartilage (n=10).
Mean values compared between condyles (Lateral/Medial; unpaired, Student's *t* test) and within subregions of condyles (Lat/Ant, Lat/Post, Med/Ant, Med/Post; *one-way* ANOVA). p-value reported.

| Cartilage Thickness [N] | | | | |
|---|---|---|---|---|
| **Condyle** | **Mean (SD)** | **Condyle** | **Mean (SD)** | **Lat - Med** |
| Lateral | 64.4 (8.4) | Medial | 46.5 (4.4) | p<0.0001 |
| *Lat/Ant* | 59.0 (8.2) | *Med/Ant* | 42.9 (5.1) | p<0.0001 |
| *Lat/Post* | 67.4 (9.7) | *Med/Post* | 49.5 (4.3) | p<0.0001 |
| Ant - Post | *ns* | Ant - Post | *ns* | - |

ns, no significant difference.
-, comparison not applicable.

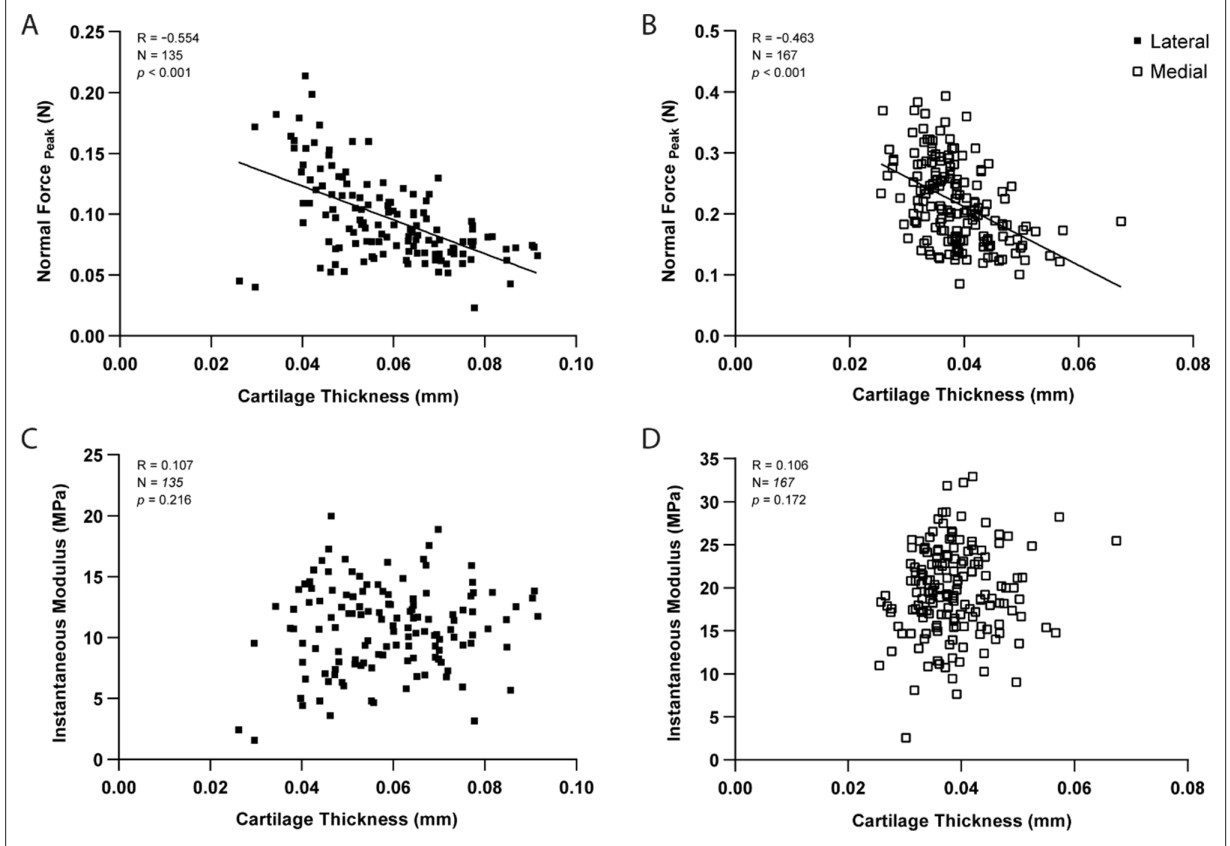

**Figure 3.** Correlation graphs per testing site for the lateral (n=135 positions) and medial (n=167) condyles, showing cartilage thickness (needle probing) is significantly correlated to peak indentation force at 20 µm (**A–B**), but not to instantaneous modulus values as determined by *Hayes et al., 1972* elastic model at 20% strain. Pearson Correlation performed and p-values reported.

These findings were supported by the combined quantitative assessment between genotypes for the different condyle regions and subregions (*Figure 4C–F*). *Prg4$^{-/-}$* mean cartilage thickness on both lateral (67.9±3.5 µm, p=0.032) and medial (56.3±3.7 µm, p<0.001) sides of the knee was higher compared to controls (*Figure 4C*). Differences in cartilage thickness between genotypes on antero-posterior sub-regions, however, were only detected on the medial condyle (*Figure 4E*). The site-specific thickness measurements enabled us to determine the instantaneous modulus at each testing site for the same mechanical strain. The utilized Hayes et al. elastic model could fit the response of mouse cartilage with great accuracy (root mean square error [RMSE] equal to 0.017±0.007 MPa for controls and 0.010±0.003 MPa for *Prg4$^{-/-}$*). Cartilage compressive stiffness was significantly lower on the medial condyle of *Prg4$^{-/-}$* mice compared to controls (8.67±1.79 MPa vs. 19.52±2.44 MPa, respectively, p<0.001). Similar patterns were seen when considering the anteroposterior subregions of the medial condyle (*Figure 4F*). No significant differences in lateral condyle cartilage stiffness were observed (10.85±1.73 MPa - control vs. 8.73±2.39 MPa - *Prg4$^{-/-}$*, p=0.187).

## Discussion

Using a novel microscale instrumented apparatus, we were able to detect and quantify spatial varia-tions in biomechanical parameters across murine cartilage surfaces, both within healthy and degener-ated femoral condyles. Compared to recent mouse studies using this commercially available apparatus (*Woods et al., 2021*; *Lavoie et al., 2015*), the optimization of sample mounting and NP protocols allowed for unprecedented quantitative mapping of the mechanical behavior and associated cartilage thickness on load-bearing regions of distal femurs with higher spatial density.

For healthy, 4-month-old C57BL/6 mice, test-retest micro-indentation measurements were repro-ducible at any given testing site, also indicating that the instantaneous deformation and sustained

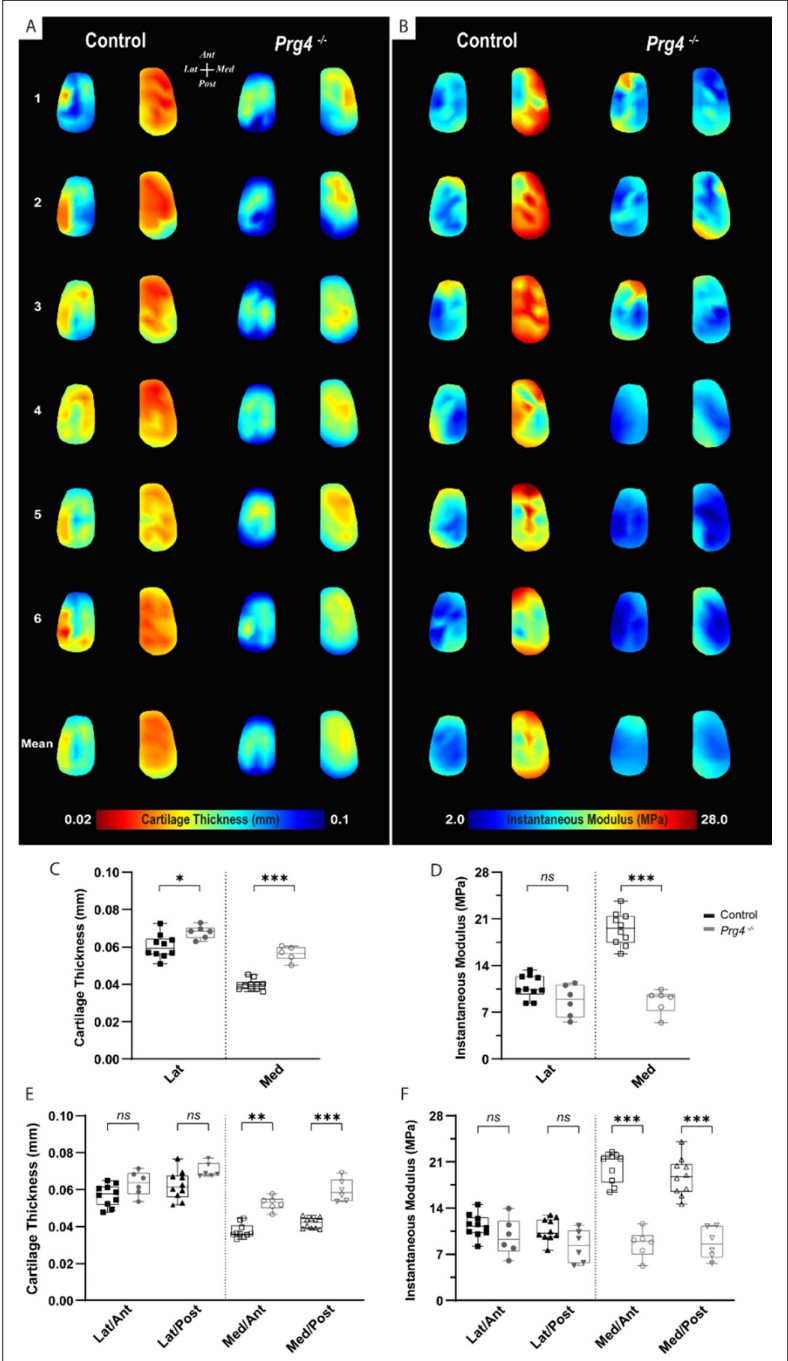

**Figure 4.** Altered biomechanical properties in degenerated murine articular cartilage. (**A–B**) Heat maps and corresponding boxplots comparing (**C–D**) regions - lateral and medial - and their (**E–F**) sub-regions - Lat/Ant, Lat/Post, Med/Ant, Med/Post - highlight spatial differences in biomechanical parameters on femoral cartilage surface between controls (C57Bl/6, n=10) and PRG4 knockout mice (Prg4$^{-/-}$, n=6). Maps shown for six representative samples per genotype, as well as the corresponding averaged map of all samples. Parameters illustrated are thickness measured by needle probing (**A, C, E**) and instantaneous modulus as determined by *Hayes et al., 1972* elastic model (**B, D, F**). Pairwise comparison between genotypes for mean values on the lateral and medial condyles and anteroposterior sub-regions (Mann-Whitney U with Bonferroni-Dunn correction, *p<0.05, **p<0.01, ***p<0.001).

The online version of this article includes the following source data for figure 4:

**Source data 1.** Instantaneous modulus used to generate data in *Figure 4*.

compression steps in the stress-relaxation protocol did not compromise the mechanical behavior of the freshly harvested cartilage tissue. Heterogeneity in peak forces was identified across anatomical locations. Notably, the medial condyle surface yielded approximately two times higher peak forces on average than its lateral counterpart. Moreover, both condyles displayed a shift toward lower values at posterior regions. Previous studies have shown significant variation in mechanical properties within different cartilage surfaces in healthy joints (*Korhonen et al., 2002*; *Froimson et al., 1997*), or even over a single surface (*Hoch et al., 1983*; *Samosky et al., 2005*). Considering the structural and material 3D complexities of articular cartilage, it is intuitive that reaction forces in indentation response are influenced by depth of indentation and applied strain rate (*Bae et al., 2006*). As thickness variations across the cartilage surfaces are unknown prior to indentation test, the relative intra-tissue strain at each testing site varied due to the fixed indentation depth imposed, thereby influencing the measured reaction forces. In general, higher peak reaction forces in intact cartilage were associated with lower thickness values. However, the effect of site-specific cartilage geometry, such as thickness, when characterizing the mechanical properties of cartilage in indentation is often overlooked (*Sim et al., 2017*; *Moshtagh et al., 2016*).

Studies in mice commonly rely on mean thickness values from histology, requiring longer turn-around and lacking spatial specificity. The optimized NP protocol used here was able to resolve spatial thickness variations across condyle surfaces and closely localizes cartilage thickness measures to the footprint of indentation on mouse distal femurs. Moreover, 3D visualization of the femoral cartilage surfaces after XRM imaging validated the positioning and distribution of the testing grid used, allowing us to pinpoint individual testing sites and compute their thicknesses. A highly significant correlation between pairwise cartilage thickness values from NP and XRM, per measurement site, was observed. Variations seen between averaged thickness values could be explained by differences in instrument resolution capabilities (Mach-1 z-axis: 0.5 µm vs XRM: 4.39 µm voxel size) and partial volume artifacts, which may affect the accuracy of tissue detection on the XRM segmentation. It is worth noting, however, that previous research (*Das Neves Borges et al., 2014*) on murine cartilage thickness has been performed using greater voxel sizes. Compromise of cartilage integrity by NP likely contributes to observed differences as well, leading to higher XRM measurements, possibly due to tissue swelling. Comparable mean cartilage thickness values between NP and XRM segmentation from contralateral femurs not subjected to biomechanical testing support this conclusion. Nevertheless, mean thickness values were in line with earlier reports for mouse cartilage (*Oláh et al., 2021*; *Poulet et al., 2013*; *Kotwal et al., 2012*), with marked greater cartilage thicknesses measured on the lateral condyle compared to the medial counterpart (*Malda et al., 2013*).

Localized cartilage thickness measurements by NP enabled us to account for the effect of cartilage geometry on indentation, with implementation of aspect ratio (indenter radius-to-cartilage-thickness) corrections as per Hayes's (*Hayes et al., 1972*) analytical formulation when calculating compressive stiffness distributions at 20% strain (considered as physiological loading; *Simha et al., 2007*). A common area of concern in in situ indentation of thin materials, such as mouse articular cartilage, is the effect of substrates on the measured mechanical properties, wherein the indentation response of articular cartilage could be somewhat influenced by the underlying rigid bone, especially on thinner areas (*Julkunen et al., 2008*). While we recognize this limitation when employing microindentation, we argue that knowledge of thickness variations improves the reliability of the indentation assessment and aids interpretation of results following degradation and loss. According to the present results, there was no significant variation in mean thickness of healthy cartilage over a condyle surface, therefore biomechanical parameters were compared between matched regions and sub-regions when assessing degeneration.

Compared to C57BL/6 control, the loss of *Prg4* function resulted in an increase in thickness and decrease in stiffness of the knee cartilage in 16-week-old mice. This is consistent with previous observations on hip articular cartilage of *Prg4*$^{-/-}$ mice (same age), using AFM indentation (*Coles et al., 2010*). Apparent modulus values in our study were about an order of magnitude larger than those reported by *Coles et al., 2010* for *Prg4*$^{-/-}$ and healthy controls, a discrepancy we attribute to differences in compression rate (1 µm/s *vs* 20 µm/s) and indentation scale (micro- *vs.* nanoindentation depth), leading to differences in strain fields, and the level of structure that is probed (*Bae et al., 2006*). It is worth noting that anatomical locations of assessment also differed between studies. The cartilage thickness values determined by NP in our study are in a similar range to those reported by *Karamchedu*

*et al., 2016*. In their micro-CT imaging study, the *Prg4*$^{-/-}$ mice displayed cartilage thickness averaging 61.0±4.3 µm and 42.4±2.6 µm for the load-bearing regions of the lateral and medial condyle, respectively. Furthermore, *Prg4*$^{-/-}$ cartilage was also shown to be significantly thicker than in control littermates (*Karamchedu et al., 2016*), even though mice were younger (10 weeks old) than in our study.

Reduction in the instantaneous compressive modulus in our study was confined to the medial compartment. We attribute that to differences in thickening in *Prg4*$^{-/-}$ femoral cartilage compared to controls, 12.6 vs 41% on lateral and medial condyles, respectively, as well as *Prg4*$^{-/-}$ compromised structural integrity (*Rhee et al., 2005*; *Jay et al., 2007*; *Drewniak et al., 2012*) having a bigger functional impact on the medial compartment, recognized as the main load-bearing region in most mammalian joints. Microscale indentation characterizes the overall tissue resistance to deformation (*Simha et al., 2007*) and disrupted architecture, and composition can impart tissue mechanical response. Due to the rapid compression rate employed in the present protocol, normal cartilage tissue is expected to deform with minimal change in volume (*Armstrong, 1986*), as its low permeability restrains fluid flow; thereby localizing the strain near the surface as fluid pressure pushes against the collagen meshwork. Thus, under instantaneous (rapid) loading the ability of cartilage to resist compression is known to be affected by the collagen fibril meshwork (*Julkunen et al., 2008*; *Laasanen et al., 2003*), particularly tangentially oriented collagen fibrils on the superficial tissue layer (*Korhonen et al., 2002*). In *Prg4*$^{-/-}$ mice, the normal parallel organization of collagen fibrils adjacent to the surface is known to be disrupted (*Jay et al., 2007*), likely affecting fluid pressurization, helping explain the lower instantaneous modulus compared to controls.

Limitations of this study include evaluation of the anatomy of the distal murine femoral condyles and not the opposing articulation (i.e. proximal tibia). Also, we focused on a single timepoint of analysis; however, investigations related to aging and progressive degeneration are of interest as they are known to affect the biochemical composition and structural integrity of cartilage (*Rhee et al., 2005*; *Rahmati et al., 2017*; *Julkunen et al., 2009a*), thereby influencing its mechanical properties. Finally, due to the biphasic/poroviscoelastic nature of cartilage, future studies should consider the use of more complex analytical models (*Mow et al., 1980*; *Julkunen et al., 2009b*) that are able to capture the time-dependent viscoelastic behavior of cartilage in mice.

In this study, we have gained insights into the patterns of varying surface geometry and mechanics present within murine articular cartilage at the microscale. 3D indentation mapping was able to resolve site-specific differences in thickness and mechanical properties across knee cartilage surfaces in healthy mice. Moreover, it identified functional changes on the *Prg4*$^{-/-}$ mouse model. This technique could also prove helpful for the study of other mouse models mimicking different mechanisms of, or therapies focused on, repair and degeneration of articular cartilage, as microscale indentation with high spatial density can provide a more comprehensive characterization of cartilage's mechanical properties.

## Materials and methods

### Animals

Male C57BL/6 mice were purchased from Jackson Laboratories (Bar Harbor, ME). Animals were housed under a standard light cycle and had free access to feed (standard diet) and water. Ten mice were euthanized at 16 weeks of age, and hind limbs (n=10 right and n=5 left) were harvested for biomechanical testing and/or 3D XRM imaging. Age-matched PRG4 knockout mice (*Prg4*$^{-/-}$, n=6) were generated and maintained on a C57BL/6genetic background, as previously described (*Abubacker et al., 2019*). Limbs were disarticulated at the hip, followed by transection of the ligaments and careful isolation of distal femurs from tibiae and menisci with the help of a dissection microscope (*Leica*). Femurs were preserved gently wrapped in Kimwipe soaked in phosphate-buffered saline (PBS, pH 7.4) until the time of assessment. All samples were mechanically tested no longer than 3 hr after dissection to prevent tissue degradation. Contralateral legs, not subjected to biomechanical testing were preserved in 10% neutral buffered formalin (NBF) solution for 24 hr and stored in 70% ethanol for 24hr to 48 hr before imaging.

### Automated indentation mapping

The shafts of isolated femurs were glued into a 0.1–10 µL pipette tip (VWR, USA) using cyanoacrylate adhesive, fixed into a stainless-steel hex nut (Paulin, Model 848–216) and secured to the

sample holder (*Figure 1—figure supplement 1*). This customized setup allowed for simple and proper positioning of the sample, exposing the load-bearing region of the condyles (*Jia et al., 2018*) for data acquisition, as well as non-destructive retrieval of samples after testing, such that subsequent XRM imaging could be carried out. A standardized mapping grid (n=31 positions) was superimposed on an image of the cartilage surface, consisting of 14 and 17 measurement sites at the lateral and medial condyles, respectively (*Figure 1A*). The testing chamber was filled with PBS solution at room temperature, and the tissue was allowed to equilibrate before testing. Automated indentation mapping under stress-relaxation was then performed using the Mach-1 v500css (Biomomentum Inc, Laval, QC) device, equipped with a calibrated multiple-axis load cell (±17 N, 3.5-mN force resolution) and associated software. At each testing site (XY coordinates), height and surface orientation were identified using four surrounding contact coordinates (front, back, left, and right) in a 0.075-mm scanning grid. Then, by concurrently moving Mach-1 stages in all three-axis at different speeds, indentation can be performed along a virtual axis normal to the surface. As per manufacturer's recommendation, contact regions for surface angles ≥60° were considered unreliable and thus discarded. In this study, a spherical indenter (0.3 mm in diameter) was driven into the cartilage to a depth of 20 µm over 1 s followed by a 90-s hold-time. For C57BL/6, a total of three indentation mappings were performed per sample, approximately 45 min apart. Data reported consist of peak force and instantaneous modulus, as determined by fitting the *Hayes et al., 1972* elastic model to the load-displacement curves at 20% strain. Assessment of how well the model fit the resulting curve per test site was done using RMSE. Since the analysis per position across specimens and genotypes reflects deviations due to anatomical positioning, calculated parameters were compared between lateral (Lat) and medial (Med) femoral condyles, as well as on four condylar sub-regions (Lateral/Anterior - Lat/Ant, Lateral/Posterior – Lat/Post, Medial/Anterior - Med/Ant, and Medial/Posterior - Med/Post), each containing at least five positions of measurement (*Figure 1A*).

## Needle probing – thickness measurement

After indentation mapping, the spherical indenter was replaced by a 30Gx1.4" hypodermic needle (TSK Laboratory, Japan) adapted to the 1-mm spherical indenter (Biomomentum Inc, Laval, Canada) (*Figure 2—figure supplement 1*). Automated thickness mapping (*Figure 2A*) was performed on the same knee cartilage surface using the NP technique (*Jurvelin et al., 1995*), thickness was measured on 31 sites located adjacent to those previously identified for the indentation mapping, while keeping the relative distance between testing sites (or overall distribution grid) the same. The needle was driven vertically into the cartilage surface at a constant speed until a 0.5-N stop criteria was reached in the subchondral bone. The cartilage surface and cartilage/subchondral bone interface positions were identified in the load-displacement curves (*Figure 2B*) generated at each measurement site using the automatic mode of analysis (*Biomomentum, 2020*). A 0.25-N/s loading limit was defined to identify the interface position. Manual correction was employed when the algorithm failed to identify the inflection point (*Biomomentum, 2020*). The cartilage thickness reported corresponds to the vertical needle displacement from cartilage surface to subchondral bone multiplied by the cosine of the surface orientation angle (*Figure 2B*) as determined during automated indentation for each position. After testing, samples were preserved in 10% NBF solution for 24 hr and stored in 70% ethanol for 24hr to 48 hr before imaging.

## 3D XRM imaging

3D XRM imaging was used for non-destructive assessment of cartilage morphology. Fixed femurs were incubated for 16–18 hr in 1% phosphotungstic acid (PTA) solution at room temperature for cartilage contrast enhancement before imaging (*Das Neves Borges et al., 2014*). Samples were enclosed onto a custom specimen chamber, with 1% PTA in 70% ethanol added to the chamber's bottom to minimize tissue dehydration. Zeiss Xradia 520 versa (Carl Zeiss X-Ray Microscopy, Pleasanton, CA) scans of each distal femur were obtained following previously described protocol (*Jablonski et al., 2021*; *Richard et al., 2020*). In brief, high-resolution scans of 2001 axial slices were acquired at a 4.39-µm voxel size, with low-energy (40 kVp voltage, 3 W power) x-rays.

## Imaging processing

The contrast-enhanced cartilage surface was segmented by determining a threshold intensity, thereby delineating the femur scan into cartilage and subchondral bone voxels. For contralateral distal femurs (n=5) that did not undergo biomechanical testing, a connected components filter was used to separate the medial and lateral condyles. The thickness transform was computed for each condyle, and the mean thickness was taken as the statistic of the thickness distribution (*Hildebrand and Rüegsegger, 1997*). For right hindlimbs, subjected to biomechanical testing, NP left a physical deformity in the articular cartilage of the right femurs visible on XRM imaging (*Figure 2A*), allowing for all 31 ROIs corresponding to NP to be manually landmarked. Landmarks were placed manually using the two-dimensional axial, sagittal, and coronal planes centered along the cartilage thickness and within each NP site. Cartilage segmentations were corrected manually to ensure the cartilage mask encompassed resulting volume gaps at NP positions. Then, the segmented cartilage was masked by a sphere of radius 75 µm placed on each landmark, leaving a thin disk of cartilage. The thickness transform was computed for each disk (*Hildebrand and Rüegsegger, 1997*) and the mean thickness values, taken as a statistic of the thickness distribution, were used to minimize variability due to morphological changes in the cartilage caused by the mechanical testing. Image processing was performed in SimpleITK (*Lowekamp et al., 2013*; Insight Software Consortium, v1.2.4), and morphometry was performed in Image Processing Language (IPL v5.42, SCANCO Medical AG, Brüttisellen, Switzerland).

## Statistical analysis

Analyses were performed in GraphPad Prism software (version 9), $\alpha$=0.05 was considered statistically significant. Continuous parameters are reported as mean values and corresponding standard deviations (SDs). Normality was assessed by Shapiro-Wilk normality test. C57BL/6 peak load and thickness data were analyzed by Student *t* test or ANOVA, with Bonferroni post hoc comparisons. To assess differences in biomechanical properties between genotypes non-parametric Mann-Whitney U test with *Bonferroni-Dunn's* correction was used. To test reliability and absolute agreement between repeated measurements, SPSS 27 (*IBM*, Chicago, IL) was used to obtain single-measurement, two-way mixed effect intraclass correlation coefficient estimates and respective 95% limits of agreement (*Koo and Li, 2016*; *Müller and Büttner, 1994*). When assessing cartilage thickness measurements between NP and XRM imaging techniques, *Pearson* correlation coefficient (R) was used for method-comparison and Bland-Altman analysis to assess bias between methods.

## Acknowledgements

Graphical representation of experimental designs was created using BioRender.com. The authors thank Biomomentum inc for their assistance during optimization of indentation testing protocol. AOM would like to thank the University of Calgary and Alberta Innovates for support during the period this study was conducted.

## Additional information

### Funding

| Funder | Grant reference number | Author |
| --- | --- | --- |
| Natural Sciences and Engineering Research Council of Canada | | Roman J Krawetz |

The funders had no role in study design, data collection and interpretation, or the decision to submit the work for publication.

### Author contributions

Anand O Masson, Conceptualization, Formal analysis, Validation, Investigation, Visualization, Methodology, Writing - original draft, Writing – review and editing; Bryce Besler, Formal analysis, Writing – review and editing; W Brent Edwards, Conceptualization, Formal analysis, Investigation, Methodology,

Writing – review and editing; Roman J Krawetz, Conceptualization, Supervision, Funding acquisition, Project administration, Writing – review and editing

### Author ORCIDs
Anand O Masson ⓘ http://orcid.org/0000-0001-7342-7038
Roman J Krawetz ⓘ http://orcid.org/0000-0002-2576-4504

### Ethics

All mouse experiments were carried out following the Canadian Council on Animal Care Guidelines recommendations and approved by the University of Calgary Animal Care Committee (protocols AC16-0043 and AC20-0042).

### Decision letter and Author response

Decision letter https://doi.org/10.7554/eLife.74664.sa1
Author response https://doi.org/10.7554/eLife.74664.sa2

## Additional files

### Supplementary files

• Transparent reporting form

### Data availability

All data generated or analyzed during this study are included in the manuscript and supporting file.

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
