## [Editor Report]

The manuscript provides supportive evidence for high-resolution analysis using indentation mapping that differentiates the biomechanical properties of normal vs. degenerated mouse articular cartilage. This study will provide the basis for future analyses for assessing degenerated articular cartilage in different mouse models (genetically engineered or assessing the therapeutic targets).

---

## [Decision Letter]

**Decision letter after peer review:**

[Editors’ note: the authors submitted for reconsideration following the decision after peer review. What follows is the decision letter after the first round of review.]

Thank you for submitting the paper "High spatial resolution analysis using indentation mapping differentiates biomechanical properties of normal vs. degenerated mouse articular cartilage" for consideration by *eLife*. Your article has been reviewed by 2 peer reviewers, and the evaluation has been overseen by a Reviewing Editor and a Senior Editor. The reviewers have opted to remain anonymous.

Comments to the Authors:

We are sorry to say that, after consultation with the reviewers, we have decided that this work will not be considered further for publication by *eLife*.

*Reviewer #1 (Recommendations for the authors):*

Needle probing underestimated cartilage thickness compared to XRM in the medial condyle.

Related to indentation thickness and substrate effects: Please acknowledge this phenomenon and pitfalls associated with mechanical indentation of thin materials in the discussion. Additional "ground truth" indentation experiments would help with the validity of this work; e.g., testing of varied thickness uniform and well-characterized material-substrate analogs (e.g., PDMS on glass) could be used to validate the indentation results of biological tissues.

Related to surface congruency: It would be compare individual 3D surface maps before and after needle probing to make sure that the indentation test does not lead to permanent deformation of the cartilage prior to needle probing. If it does, considerations can and should be made regarding propagated error of this technique.

RSME should be corrected on lines 403 to RMSE.

*Reviewer #2 (Recommendations for the authors):*

This study uses a commercial instrument to map the mechanical properties of cartilage in the mouse knee. Many claims are made regarding the novelty of the work but these are generally minor advances and the claims are overstated. Overall there are major fundamental limitations with the method being used. The validation of the needle-probe thickness measure is a novel aspect of the work.

1. In many cases, it is noted that certain aspects of the studies have not been performed previously, but the overall rationale and motivation for doing so is missing. Thus it is not clear how this work is truly different than several previous studies. Primary example is " This apparatus has been recently employed by Woods and colleagues to evaluate altered biomechanics in a mouse model of cartilage degeneration. However, the study was conducted in the non-load-bearing region of the mouse knee joint." The advance is to apply this technique to a different region of the joint, and to a mouse model that has already been shown to show degenerative changes.

2. Much of the reporting focuses on peak force. This value is highly dependent on indenter size, rate of loading, etc. and thus is not meaningful, as compared to intrinsic cartilage properties.

3. The authors comment that previous studies on indentation mapping have not measured its accuracy. This is not possible, nor was the accuracy of the modulus measurements made here. In general, many of the claims about novelty of this study are overstated.

4. There are several technical issues. For example, the surface of the joint is curved yet indentation is not adjusted to be performed perpendicular to the surface. This can have a significant effect on the measurements. Was the "shift" in values in the posterior area corrected for this curvature? What was the contact region of the indenter at these angles of indentation?

5. The study focuses on instantaneous modulus, which is only a single measure of cartilage properties and cannot be used in predictive modeling. There is no comparison of the current findings to those of previous mouse indentation modulus values. In fact, the numbers reported here seem to be an order of magnitude higher than previous reports.

6. The modulus of Prg4 mouse cartilage was reported already (although not in site-specific manner). Again the claims of novelty are overstated and no comparison is made to previous reports.

7. How is statistical analysis handled for comparison of so many values? It is not clear that the proper corrections are made for multivariate analysis at all the different sites.

---

## [Author Response]

[Editors’ note: The authors appealed the original decision. What follows is the authors’ response to the first round of review.]

Reviewer #1 (Recommendations for the authors):Needle probing underestimated cartilage thickness compared to XRM in the medial condyle.

It is possible that needle probing underestimated cartilage thickness at thinner regions. However, we would like to point out that the average difference between cartilage thickness measurements for the medial condyle ranged from 5.5 to 7.8 µm, representing 1.25 to 1.77-pixel difference. The XRM resolution (4.4 µm/pixel) allowed for eight to ten pixels across the cartilage thickness, which may affect the accuracy of tissue detection. Partial volume artifacts could further affect the segmentation, but previous research [1] has been performed using pixel sizes greater (5 μm/pixel, for example) than those used in our work.

Based on our existing and new data, we stand by our previous claim that resolution and partial volume artifacts are confounding factors on comparisons between NP and XRM point-specific thickness measurements as, proportionally, these would have a bigger impact on thinner regions. Compromise of cartilage integrity by needle probing could have also contributed to differences in point-specific thickness values between methodologies (likely leading to overestimation by XRM segmentation due to tissue swelling and partial volume artifacts). Thus, we completed image processing of previously harvested and imaged contralateral legs (*n*=5) of the same C57Bl6 mice that had not undergone mechanical testing (i.e., indentation/needle probing). The resulting heatmaps of cartilage thickness distributions across femoral condyles pointed to thicker cartilage on the lateral condyle, with mean cartilage thicknesses of 40.63 ± 2.14 µm and 56.99 ± 6.26 μm for the medial and lateral condyle surfaces, respectively. These values were in line with mean thickness values retrieved from NP (Table 2) and XRM post-NP (Table S2). We added two sentences regarding these results, and the image below was added to manuscript as supplementary Figure S3.

Related to indentation thickness and substrate effects: Please acknowledge this phenomenon and pitfalls associated with mechanical indentation of thin materials in the discussion. Additional "ground truth" indentation experiments would help with the validity of this work; e.g., testing of varied thickness uniform and well-characterized material-substrate analogs (e.g., PDMS on glass) could be used to validate the indentation results of biological tissues.

We thank the reviewer for this valuable suggestion, and we agree that it would be constructive to carry out this analysis, which can be completed in 2 to 3 weeks using standard materials with similar material properties to cartilage and bone.

To investigate the possible effects of the underlying bone on indentation response (substrate effect), silicone samples of 0.5 mm thickness were attached to synthetic bone analogs of different moduli (Author response image 1). Blocks of rigid polyurethane foam PCF5 and PCF50 (Sawbone, Maomoe, Sweden) with 1148MPa and 16MPa compression modulus (ASTM D1621) respectively, where cut into 2cm x 2cm x 1cm thick samples prior to testing. Indentation was performed across five adjacent testing sites using two different size indenters (0.3mm and 1.0 mm in diameter) up to 40% strain. Load-displacement data was reproducible across indentation sites. Rise in reaction forces for bigger probing size (comparing Author response image 1 and 1C for same substrate type) are a result of different contact areas and intra-specimen stress-strains. A 2-to-3.33-fold increase in peak force for same indentation depth and substrate, however, corresponded to ≤ 2MPa difference in instantaneous modulus. As for substrate effects, a 7000% increase in substrate modulus (1148MPa *vs* 16Mpa) led to 1.38-to-2.29-fold increase in peak force and less than a 2-fold increase on the average instantaneous modulus at 20% strain, for both indenter sizes*.* Thus, indentation response is somewhat influenced by indenter-radius-to-sample-thickness ratio (aspect ratio) and substrate, with increasing reaction forces, but Hayes *et al.* (1972) scaling factor correction reduces sensitivity of derived elastic modulus to sample thickness (aspect ratio) and substrate material contributions. The average instantaneous modulus retrieved for PCF50 under indentation was 850.9 ± 23.8 MPa (0.3 indenter) and 885.4 ± 48.6 Mpa (1.0 indenter). The difference in measured and reported modulus are due to test differences, indentation versus uniaxial compression, respectively. In this study, a strain higher than 0.2% could not be achieved for this rigid material because the maximum force criteria set as per manufacturer recommendation (to avoid small indenter tip damage) was reached with minimal deformation.

**Author response image 1. sa2fig1:** (**A**) Indentation protocol for substrate effect testing, at an indentation rate of 0.1mm/s, at a maximum 40% strain. Silicone samples of 0.5 mm thickness were attached to synthetic bone analogs of different moduli – PCF5 and PCF50 (Sawbone, Sweden) and indentation response across five adjacent sites was evaluated using the load-displacement curves for a (**B**) 0.3mm indenter and (**C**) 1.0mm indenter testing and (**D**) instantaneous elastic modulus as determined by Hayes et al. (1972) elastic model at 20% strain.

Author response image 1 has not been added to the manuscript. However, we would be happy to include it as a supplementary figure if reviewers deem it necessary.

Related to surface congruency: It would be compare individual 3D surface maps before and after needle probing to make sure that the indentation test does not lead to permanent deformation of the cartilage prior to needle probing. If it does, considerations can and should be made regarding propagated error of this technique.

We edited the Methods section to help elucidate this matter and added to the sentence that needle probing measurements were taken at adjacent positions to indentation whilst keeping the distribution grid (relative distance between positions) the same. We reiterate that the demonstrated test-retest repeatability in indentation peak forces and load-displacement curves suggests indentations did not damage the cartilage surfaces prior to needle probing. This can be further corroborated by close inspection of 3D PTA-XRM images (Author response mage 2), wherein no surface changes indicative of permanent deformation was observed on femurs subjected to indentation testing only (A), nor adjacent to the clearly visible needle probing sites on samples that underwent indentation followed by needle probing (B). Figure 2 has been updated in the manuscript by adding the zoomed in image displaying needle probing markings (A).

**Author response image 2. sa2fig2:** Representative 3D PTA-XRM volume renderings of distal femoral condyles subjected to indentation testing (A) and indentation followed by needle probing (B).

RSME should be corrected on lines 403 to RMSE.

This has been corrected.

Reviewer #2 (Recommendations for the authors):This study uses a commercial instrument to map the mechanical properties of cartilage in the mouse knee. Many claims are made regarding the novelty of the work but these are generally minor advances and the claims are overstated. Overall there are major fundamental limitations with the method being used. The validation of the needle-probe thickness measure is a novel aspect of the work.1. In many cases, it is noted that certain aspects of the studies have not been performed previously, but the overall rationale and motivation for doing so is missing. Thus it is not clear how this work is truly different than several previous studies. Primary example is " This apparatus has been recently employed by Woods and colleagues to evaluate altered biomechanics in a mouse model of cartilage degeneration. However, the study was conducted in the non-load-bearing region of the mouse knee joint." The advance is to apply this technique to a different region of the joint, and to a mouse model that has already been shown to show degenerative changes.

While we respect the reviewers opinion, we do not agree with their interpretation of the novelty or importance/impact of our study. This study is novel and impactful because of the following reasons:

It is the first to map biomechanical properties at multiple, spatially distributed testing sites across murine femoral condyle surfaces and on the load-bearing part of the joint [The study indicated by the reviewer – Woods et al., was undertaken in a region of the joint that is non-load bearing in the mouse and therefore has minimal physiological relevance].Despite the use of this commercially available mechanical apparatus in previous studies, no one has yet to report on the precision and repeatability of indentation measurements for mouse cartilage has been previously demonstrated. Therefore, this is an essential finding to allow researchers/the field to have confidence in the methodology.Needle probing cartilage mapping has never been successfully undertaken in mice, and as the reviewer stated themselves, this is the first time point-by-point validation of cartilage thickness measurements has been performed, as opposed to mean thickness comparisons that the field has relied on up to now.We decided to use popular mouse model of reproducible cartilage degeneration in the study – the Prg4 knockout because of two main reasons. The first is that it is spontaneous and doesn’t rely on injury/intervention as these types of models have shown various differences in disease trajectories and cartilage degeneration depending on the lab performing the experiments. The second reason, is that the cartilage mechanical properties of the Prg4 knockout have already been reported in the literature and this is essential for the validation of the methodology – it is essential to compare our findings with that of a reproducible, spontaneous model of cartilage degeneration with minimal bias that could be introduced into the system by the ones performing the research. Our method not only showed spatial variation in stiffness and thickness for the degenerated cartilage in Prg4 knockout mouse, but was also comparable to previously published literature. This is a considerable strength in our experimental design and not a weakness as the reviewer believes.

We added a statement about the benefits of using spontaneous models of cartilage degeneration, such as *Prg4* knockout mice, to corroborate the use of this mouse model when validating the methodology employed in our study. Based on reviewer’s comments we also expanded on comparisons of current findings to previous studies (see response to comments 5 below).

2. Much of the reporting focuses on peak force. This value is highly dependent on indenter size, rate of loading, etc. and thus is not meaningful, as compared to intrinsic cartilage properties.

We agree that peak force is dependent on indenter size and other indentation testing parameters. That being said, reporting focused on peak force was used to investigate test-retest reliability of the mechanical apparatus regarding murine cartilage measurements. To that extent, raw data is more compelling than processed data, which in the end is required for calculation of intrinsic cartilage properties anyways.

3. The authors comment that previous studies on indentation mapping have not measured its accuracy. This is not possible, nor was the accuracy of the modulus measurements made here. In general, many of the claims about novelty of this study are overstated.

We appreciate the reviewer’s comment and would like to rectify the use of the term “accuracy”. Previous studies on mouse cartilage indentation mapping have not demonstrated the precision and repeatability of indentation measurements from the apparatus used.

We edited the manuscript clarifying that our study investigated the precision and test-retest reliability, not accuracy, of the proposed automated indentation mapping technique for mouse articular cartilage.

4. There are several technical issues. For example, the surface of the joint is curved yet indentation is not adjusted to be performed perpendicular to the surface. This can have a significant effect on the measurements. Was the "shift" in values in the posterior area corrected for this curvature? What was the contact region of the indenter at these angles of indentation?

It appears that the reviewer misunderstood the methodology used in the paper and we would be happy to make it clearer in the methods section, however this comment is incorrect. We would like to clarify that Biomomentum’s mechanical tester was developed to overcome such limitations. Specifically, it detects the height and orientation of the surface at the XY position using four adjacent contact coordinates to each testing site (front, back, left, and right) in a 0.075mm scanning. Then, indentation is performed by simultaneously moving the three stages of the tester at different speeds, which allows the spherical indenter to be moved along a virtual axis normal to the surface of the sample. In sum, by identifying the surface orientation, the apparatus can correct for surface curvature before indentation measurement allowing for testing of curved surface, like the joints. Contact regions for surface angles ≥ 60^o^ are unreliable and thus discarded.

We edited the Methods section in the manuscript and clarified how the Mach-1 mechanical tester is capable of identifying surface orientation prior to indentation measurement and thus, perform indentation normal to the surface of angled samples (like the femoral condyle). Of note, similar description highlighting such capabilities and its advantage had already been reported on the manuscript’s introduction (lines 69-76).

5. The study focuses on instantaneous modulus, which is only a single measure of cartilage properties and cannot be used in predictive modeling. There is no comparison of the current findings to those of previous mouse indentation modulus values. In fact, the numbers reported here seem to be an order of magnitude higher than previous reports.

We thank the reviewer for this valuable comment and would be happy to include further comparisons to previous studies in the field. Indeed, given cartilage’s poro-viscoelastic nature, its compressive modulus magnitude can be affected by the rate of loading to which it is subjected. It has already been demonstrated that the compressive modulus determined at shorter loading times (instantaneous compression), can be greater than modulus retrieved at lower speed indentations. We also used freshly harvested rather than frozen-thawed samples.

We added a couple sentences addressing comparisons to other studies and differences between studies that can explain such discrepancy in indentation values, as abovementioned.

6. The modulus of Prg4 mouse cartilage was reported already (although not in site-specific manner). Again the claims of novelty are overstated and no comparison is made to previous reports.

This is the first time microscale automated indentation combined with needle probing mapping has been used to identify regional changes on Proteoglycan 4 knockout mice distal femurs. Previous studies using the same mouse strain, employed different techniques (AFM, Histology, microCT) and/or assessed other joint tissues (hip, ankle). Changes in thickness and stiffness in Prg4 knockout mice were in line with these studies and we would be happy to further expand on comparisons with previous reports.

7. How is statistical analysis handled for comparison of so many values? It is not clear that the proper corrections are made for multivariate analysis at all the different sites.

Post hoc test was used for multiple comparison correction, and this was done up to the standard of what is published in the field. However, we would be happy to further explain it in the methods section.

[1] Das Neves Borges, P., Forte, A. E., Vincent, T. L., Dini, D. & Marenzana, M. Rapid, automated imaging of mouse articular cartilage by microCT for early detection of osteoarthritis and finite element modelling of joint mechanics. *Osteoarthr. Cartil.* 22, 1419–522 1428 (2014)